# The molecular transition that confers voltage dependence to muscle contraction

Marina Angelini [1] ✉, Nicoletta Savalli[1], Federica Steccanella[1], Savana Maxfield[1], Serena Pozzi[2], Marino DiFranco[3], Stephen C. Cannon [3], Antonios Pantazis [2,4] & Riccardo Olcese [1,3] ✉

What is the molecular origin of voltage dependence in skeletal muscle excitation-contraction? Cholinergic transmission to the muscle fiber triggers action potentials, which are sensed by voltage-gated L-type calcium channels ($Ca_V1.1$). In turn, the conformational changes in $Ca_V1.1$ propagate to and activate intracellular ryanodine receptors (RyR1), causing $Ca^{2+}$ release and contraction. The $Ca_V1.1$ channel has four voltage-sensing domains (VSD-I to -IV) with diverse voltage-sensing properties, so the identity of VSD(s) responsible for conferring voltage dependence to RyR1 opening, is unknown. Using voltage-clamp fluorometry, we show that only VSD-III possesses kinetic, voltage-dependent and pharmacological properties consistent with skeletal-muscle excitability and $Ca^{2+}$ release. We propose that the earliest voltage-dependent event in the excitation-contraction process is the structural rearrangement of VSD-III that propagates to RyR1 to initiate $Ca^{2+}$ release and contraction.

Skeletal muscle contraction is initiated by an electrical excitation, the muscle action potential (AP), which triggers the opening of ryanodine receptors (RyR1) and the release of $Ca^{2+}$ from the sarcoplasmic reticulum (SR)[1]. This sequence of events, referred to as excitation-contraction (EC) coupling[2], relies on the physical association of RyR1, an intracellular channel that is intrinsically voltage-insensitive, with four voltage-gated $Ca^{2+}$ channels ($Ca_V1.1$, also known as the dihydropyridine receptor, DHPR) located in the sarcolemma and T-tubules (Fig. 1)[3,4].

During an action potential, the $Ca_V1.1$ channel is thought to transduce the electric potential change across the membrane of muscle fibers by undergoing conformational rearrangements that propagate to and open RyR1, causing contraction (Fig. 1)[1,5]. Early studies on muscle EC coupling recognized that an intramembrane charge movement originating from $Ca_V1.1$ associates with voltage-dependent SR $Ca^{2+}$ release[6–8]. This measurable current, called "gating current"[9], is the electrical manifestation of the concomitant activation of the four voltage-sensing domains (VSDs) of $Ca_V1.1$,[6,7,10,11] (Fig. 1

and Supplementary Fig. 1a), compelled to by a change in membrane potential. As gating currents are a composite signal generated by the concomitant activation of all four VSDs, they cannot be used to extract information about individual VSDs. Prior studies have attempted to uniquely pinpoint the specific VSD(s) that control EC coupling using different approaches in cultured murine muscle fibers and myotubes, reaching wholly contrasting conclusions[11,12]: an optical approach reported that all VSDs may be involved except VSD-III[11], while a mutagenesis approach concluded that only impairment of VSD-III affects $Ca^{2+}$ release[12].

To solve this recent conundrum and definitively answer the age-old question of how excitation and contraction are coupled, we defined and tested a series of stringent criteria that a $Ca_V1.1$ VSD coupled to RyR1 opening should satisfy:

1. VSD activation is kinetically compatible with the time courses of the muscle action potential and $Ca^{2+}$ release.
2. VSD activation and $Ca^{2+}$ release have shared voltage dependence, specifically:

[1]Division of Molecular Medicine, Department of Anesthesiology & Perioperative Medicine, David Geffen School of Medicine, University of California Los Angeles, Los Angeles, CA, USA. [2]Division of Cell and Neurobiology, Department of Biomedical and Clinical Sciences, Linköping University, Linköping, Sweden. [3]Department of Physiology, David Geffen School of Medicine, University of California Los Angeles, Los Angeles, CA, USA. [4]Wallenberg Center for Molecular Medicine, Linköping University, Linköping, Sweden. ✉e-mail: mangelini@ucla.edu; rolcese@ucla.edu

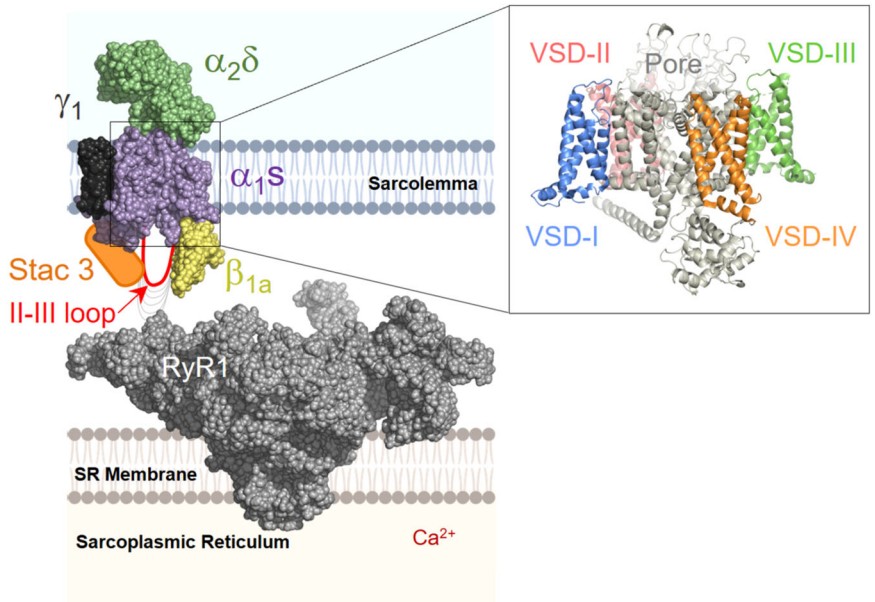

**Fig. 1 | Key players in skeletal muscle EC coupling: Ca$_V$1.1, Stac3 and RyR1.** Ca$_V$1.1 channel macromolecular complex (α$_{1S}$ pore forming subunit (violet), auxiliary subunits β$_{1a}$ (yellow), α$_2$δ−1 (green), and γ1 (black), and Stac3 adapter protein, orange) is thought to be in physical contact with RyR1 (gray). α$_{1S}$ comprises four concatenated repeats (I-IV), each of which includes a voltage-sensing domain (VSD, blue, red, green, and orange) and a quarter of the pore domain (enlargement). RyR1 is embedded in the SR membrane and cannot sense electrical signals in the sarcolemma directly; instead, it uses Ca$_V$1.1 voltage-sensing apparatus to gain voltage dependence. During an action potential, the voltage-dependent conformational rearrangements in one or more Ca$_V$1.1 VSD(s) are thought to propagate to RyR1, likely *via* the repeat II-III loop (red)[50], which opens and releases Ca$^{2+}$ from the sarcoplasmic reticulum (SR). (Ca$_V$1.1 Protein Data Bank ID 5GJV[53], RyR1 Protein Data Bank ID 3J8H[66]).

---

 a. The VSD is not active at potentials where Ca$^{2+}$ release does not occur, and

 b. The VSD activates over the range of membrane potentials where Ca$^{2+}$ release occurs.

3. A drug that modifies the activation of the VSD(s) assigned to RyR1 should also affect Ca$^{2+}$ release.

In this work, to test these postulates, we use the Cut-open Oocyte Vaseline Gap Voltage-Clamp Fluorometry (COVG-VCF) approach, which affords exceptional voltage-dependent optical tracking of protein conformational changes under physiologically relevant conditions. In this way, we resolve the activation of the four individual VSDs in the human Ca$_V$1.1 channel macromolecular complex (Fig. 1), under diverse and naturalistic voltage protocols and while modified by an antihypertensive drug.

## Results

### Only VSD-III and -IV display kinetics compatible with muscle Ca$^{2+}$ release and activate in response to a mammalian muscle action potential

COVG-VCF, a combined electrophysiological-optical approach, is able to detect the voltage and time-dependent conformational rearrangements of individual VSDs[10,13] revealing the molecular events that initiate EC coupling in skeletal muscle.

A cysteine (Cys) was introduced at the extracellular flank of S4 helices in each VSD of the Ca$_V$1.1 pore-forming subunit (α$_{1S}$) and labeled with a thiol-reactive fluorophore (Supplementary Fig. 1a) serving as an optical reporter of VSD local conformational changes. The engineered channels retained WT-like functional properties (Supplementary Fig. 1b–d and Supplementary Table 1).

Under physiological conditions (functional channels) and 2 mM external [Ca$^{2+}$], we simultaneously recorded ionic current and fluorescence changes (reporting transmembrane movements of the individual VSDs) from the entire human Ca$_V$1.1 channel expressed in *Xenopus* oocytes (Fig. 2a).

Each VSD reacted to depolarizing pulses with distinct kinetics and voltage dependence (Fig. 2a). VSD-I displayed the slowest kinetics of activation (τ activation at 20 mV = 76.9 ± 6.5 ms, $n$ = 9) while VSD-III and -IV activated very rapidly with time constants shorter than 2 ms (τ activation: VSD-III = 1.81 ± 0.2 ms, $n$ = 9, VSD-IV = 1.79 ± 0.2 ms, $n$ = 6, Fig. 2c). On the other hand, VSD-II, whose dominant time constant of activation is slow (τ$_{slow}$ = 212.1 ± 31.6 ms, 71% ± 4% of the total amplitude), displayed a minor fast component of 3.2 ± 0.6 ms ($n$ = 5) that accounted for 29% ± 4% of the total amplitude (Fig. 2c). Note that Ca$^{2+}$ release from the SR is a very fast event that peaks within 3-4 ms[11,14]: thus, in principle, VSD-II, -III and -IV possess at least one kinetic component of activation fast enough to allow their movement in response to a muscle AP (3–6 ms). This is a necessary condition to act as a voltage sensor for RyR1 activation and fast Ca$^{2+}$ release.

To determine the sensitivity of each VSD to a physiological stimulus, we optically recorded the activation of each VSD (Fig. 2b and Supplementary Movie 1) using a mammalian skeletal muscle AP waveform as a voltage command. The collected fluorescence signal integrates both voltage dependence and kinetic properties of the individual VSD and informs as to whether a specific sensor is activated at rest and by a physiological AP (duration at half-maximum amplitude = 1.5 ms). First, we found that AP-evoked conformational changes could be detected in all VSDs, similar to optical studies in cultured murine fibers[11]. Yet, stark differences in the fractional activation of each VSD were revealed when the data were normalized to the limiting maximal and minimal ΔF from the same cell. We found that at the resting membrane potential (Vm = −93 mV), the activity of VSD-IV is quite high (17% ± 2%), a feature that disqualifies this sensor from having a role in SR Ca$^{2+}$ release. On the other hand, the other VSDs had no significant activity at resting membrane potential (Fig. 2b and d). We found that only VSD-III and -IV reached a probability of activation of 58% ± 5% ($n$ = 5) and 63% ± 4% ($n$ = 6) of the maximal activation, respectively (Fig. 2b and e). Not surprisingly, the signals from the

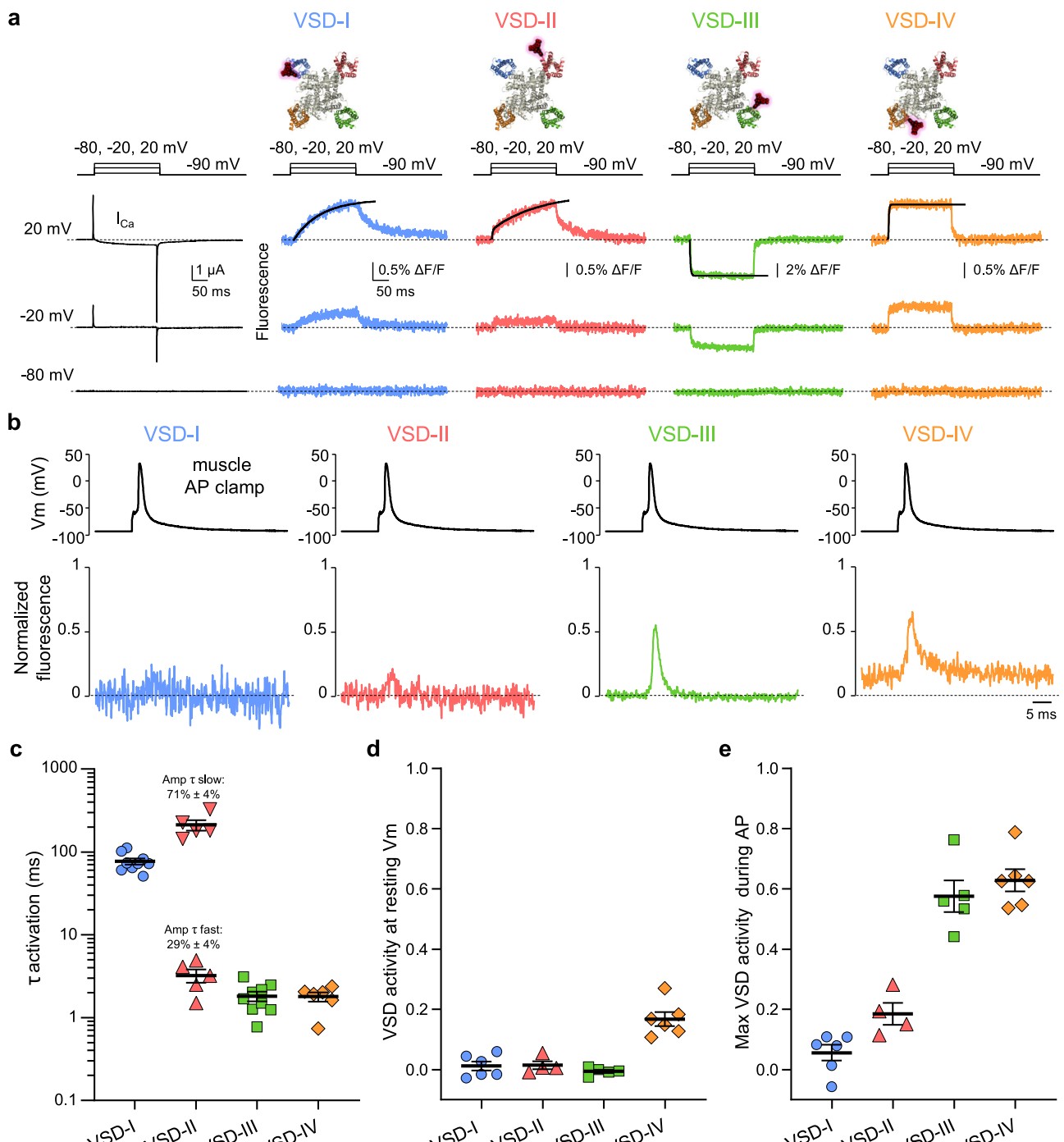

**Fig. 2 | Characterization of time and voltage-dependent properties of the four Ca_V1.1 voltage-sensing domains. a** Representative $Ca^{2+}$ currents ($I_{Ca}$) and simultaneously recorded fluorescence signals from human Ca_V1.1 channels complex ($\alpha_{1S} + \beta_{1a} + \alpha_2\delta\text{-}1 + \gamma1 + Stac3$) reporting local protein structural changes in each VSD. Above the recordings are the step voltage protocol used and the labeling position in the pore-forming $\alpha_{1S}$ subunit of Ca_V1.1 channels structure (Protein Data Bank ID 5GJV;[53], top view). The black lines superimposed to the fluorescence traces at 20 mV are best fits to a single (VSD-I, -III, and -IV) or double exponential functions (VSD-II). **b** Mean normalized fluorescence representing the VSD activation probabilities during a skeletal muscle AP waveform. For comparison, all the fluorescence traces are presented as positive deflections. An animation combining these AP-clamp data on the Ca_V1.1 structure is shown in Supplementary Movie 1. **c** The time constant ($\tau$) of VSD activation to 20 mV. For VSD-II, the relative amplitudes (Amp%) of fast and slow components are reported above the $\tau$ values (VSD-I: $n = 9$, VSD-II: $n = 5$, VSD-III: $n = 9$, VSD-IV: $n = 6$). **d** VSDs activities at resting membrane potential (Vm). Note the elevated activity of VSD-IV at resting membrane potential (VSD-I: $n = 6$, VSD-II: $n = 4$, VSD-III: $n = 5$, VSD-IV: $n = 6$). **e** Maximal VSD activity recorded during AP-clamp from experiments as in (**b**). Note that only fast-activating VSD-III and -IV respond to an AP stimulus. (VSD-I: $n = 6$, VSD-II: $n = 4$, VSD-III: $n = 5$, VSD-IV: $n = 6$). Error bars are ± SEM.

slowest sensors (VSD-I and -II) were barely detectable as they did not promptly respond to the brief AP, reaching an activation probability of only $6 \pm 3\%$, $n = 6$ (VSD-I) and $19 \pm 4\%$, $n = 4$ (VSD-II) (Fig. 2b and e).

## Only VSD-III displays both fast kinetics and voltage dependence compatible with skeletal muscle $Ca^{2+}$ release

VSD-III and -IV are able to activate in response to an AP: however, to act as bona fide RyR1 voltage sensor, a VSD needs to match the voltage

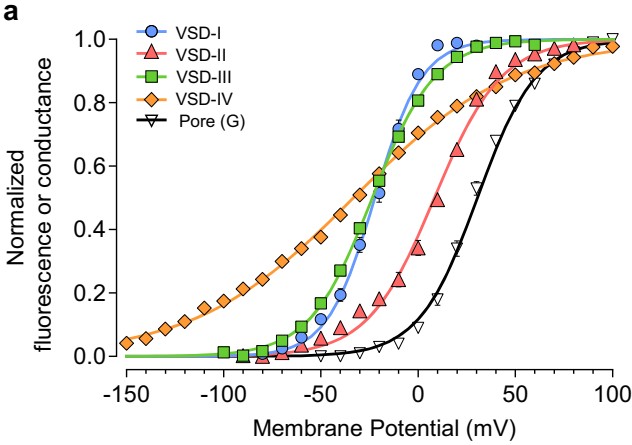

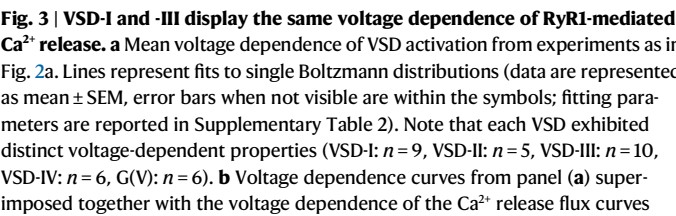

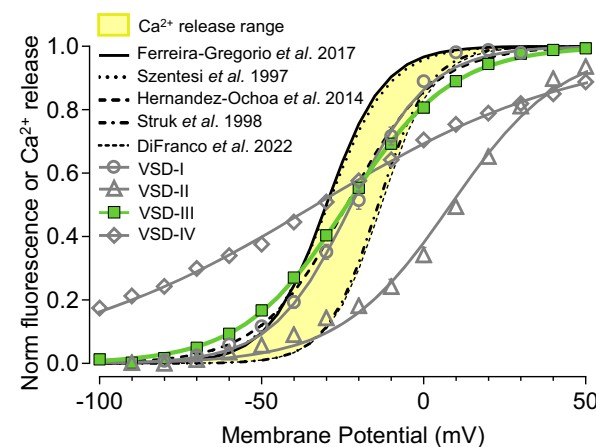

**Fig. 3 | VSD-I and -III display the same voltage dependence of RyR1-mediated Ca²⁺ release. a** Mean voltage dependence of VSD activation from experiments as in Fig. 2a. Lines represent fits to single Boltzmann distributions (data are represented as mean ± SEM, error bars when not visible are within the symbols; fitting parameters are reported in Supplementary Table 2). Note that each VSD exhibited distinct voltage-dependent properties (VSD-I: $n = 9$, VSD-II: $n = 5$, VSD-III: $n = 10$, VSD-IV: $n = 6$, G(V): $n = 6$). **b** Voltage dependence curves from panel (**a**) superimposed together with the voltage dependence of the Ca²⁺ release flux curves

(black lines) from mammalian adult skeletal muscle fibers for comparison. The reported curves (black lines) were constructed from the Boltzmann distribution fit parameters in (i) Ferreira-Gregorio et al., 2017: $V_{1/2} = -30$ mV, $z = 2.85$ $e_0$[18], (ii) Szentesi et al., 1997: $V_{1/2} = -29.5$ mV, $z = 2.7$ $e_0$[16], (iii) Hernandez-Ochoa et al., 2014: $V_{1/2} = -24$ mV, $z = 1.92$ $e_0$[17], (iv) Struk et al., 1998: $V_{1/2} = -14$ mV, $z = 3.25$ $e_0$[15], (v) DiFranco et al., 2022: $V_{1/2} = -13$ mV, $z = 3.13$ $e_0$[19]. The yellow highlighted area delimits the ranges of membrane potentials where the Ca²⁺ release flux occurs. Note that only VSD-I and -III activate with a voltage dependence similar to that of SR Ca²⁺ release.

dependence of RyR1-mediated Ca²⁺ release, a measurable parameter in muscle EC coupling. To identify the VSDs with a voltage dependence compatible with that of skeletal muscle Ca²⁺ release, we constructed their voltage-dependent activation curves. As shown in Fig. 3a, the four VSDs are highly heterogeneous in their voltage sensitivity ($z$, effective valence) and potential of half-activation ($V_{1/2}$) (Fig. 3a and Supplementary Table 2):

VSD-IV displayed the most negative half-activation potential and was the least sensitive to changes in membrane potential ($V_{1/2} = -32 \pm 2$ mV; $z = 0.6 \pm 0.02$ $e_0$, $n = 6$). VSD-II moves at the most depolarized membrane potentials with a $V_{1/2} = 8 \pm 1$ mV, and $z = 1.5 \pm 0.1$ $e_0$ ($n = 5$). VSD-I and VSD-III exhibited similar activation (VSD-I: $V_{1/2} = -22 \pm 1$ mV; $z = 2.2 \pm 0.1$ $e_0$ $n = 9$; VSD-III: $V_{1/2} = -24 \pm 1$ mV; $z = 1.6 \pm 0.04$ $e_0$, $n = 10$) (Fig. 3a and Supplementary Table 2).

Remarkably, VSD-I and VSD-III display an overall voltage dependence that closely follows that reported for Ca²⁺ release in muscle fibers measured in 1.8-2.0 mM [Ca²⁺]ₑₓₜ as in the present study. Figure 3b illustrates this finding, showing the activation curves (F(V)) of the four VSDs superimposed with Ca²⁺ release curves obtained from human[15] and rodent[16–19] studies of adult muscle fibers. Note that only VSD-I and -III voltage dependence falls in the narrow membrane potential range where SR Ca²⁺ release occurs. Importantly, these results dismiss VSD-II and -IV as sensors for RyR1 activation (and Ca²⁺ release).

Thus, VSD-III stands out as the single sensor with kinetics (Fig. 2) and voltage dependence (Fig. 3) of the skeletal muscle Ca²⁺ release process.

**Pharmacological evidence reinforces VSD-III role as the voltage sensor of muscle contraction**
Dihydropyridines (DHPs) are a class of Ca²⁺ channel blockers that bind to the pore-forming subunit of L-type Caᵥ channels, including Caᵥ1.1. Nifedipine, a well-known antihypertensive and antianginal drug[20], binds to the pore region at the interface of Repeats III and IV[21–23] (Fig. 4a), preventing Ca²⁺ entry (Fig. 4b, c). Paradoxically, although Ca²⁺ influx through the Caᵥ1.1 is not required for skeletal muscle contraction[24,25], numerous studies have found that this drug alters, SR Ca²⁺ release and contraction in mammalian and amphibian muscles[7,25–39]. Remarkably, nifedipine was found to perturb the charge movement of Caᵥ1.1[7,30]. Together, this body of literature

provides strong evidence that nifedipine modifies the behavior of the VSD(s) that confer voltage dependence to RyR1 opening and RyR1-mediated SR Ca²⁺ release. Figure 4 shows representative fluorescence recordings in control (no drug) and in the presence of 10 μM nifedipine. Note that nifedipine binding to the pore perturbed VSD-III activation, such that its voltage dependence shifted towards hyperpolarized potential by ~20 mV (control: $V_{1/2} = -25 \pm 1$ mV, $n = 14$; nifedipine $V_{1/2} = -43 \pm 2$ mV, $n = 14$, $P < 0.001$). To a lesser extent, also the activation of VSD-I was affected (control: $V_{1/2} = -30 \pm 2$ mV, $n = 11$, nifedipine $V_{1/2} = -45 \pm 2$ mV, $n = 5$, $P < 0.001$). On the other hand, nifedipine did not modify VSD-II and -IV (Fig. 4f and Supplementary Table 3).

In contrast to VSD-I, VSD-III activation is not strongly coupled with the pore-domain open conformation, in the absence of nifedipine[10]. Moreover, a nifedipine-bound channel structure did not show nifedipine binding on VSD-III[23]. How does nifedipine facilitate VSD-III activation? One straightforward explanation is that the pore structural configuration induced by nifedipine binding is conformationally coupled to both VSD-I and -III, such that the activations of both domains are allosterically facilitated. In related L-type Caᵥ1.2 channels, VSD-III and the pore are conformationally coupled[40], so nifedipine binding to Caᵥ1.1 may not need to induce a very drastic structural change to influence VSD-III.

In conclusion, these results further demonstrate the relevance of VSD-III to skeletal muscle EC-coupling, since this is the sole sensor that: (i) possesses fast kinetics compatible with that of Ca²⁺ release, (ii) shares the same voltage dependence as of SR Ca²⁺ release and (iii) is perturbed by nifedipine, a calcium channel blocker that alters Ca²⁺ release in skeletal muscle (Fig. 4f).

## Discussion
### VSD-III of Caᵥ1.1 is the voltage sensor of SR Ca²⁺ release
In 1791, Italian scientist Luigi Galvani made a groundbreaking observation, associating for the first time muscle contraction with electricity[41]. Since then, great progress has been made in identifying the players of the signaling cascade that, starting from the electrical excitation, leads to contraction[11,42–45]. Remarkably, Schneider and Chandler recognized that a movement of charge within the sarcolemma was a key step in skeletal muscle EC coupling[6]. A few years later,

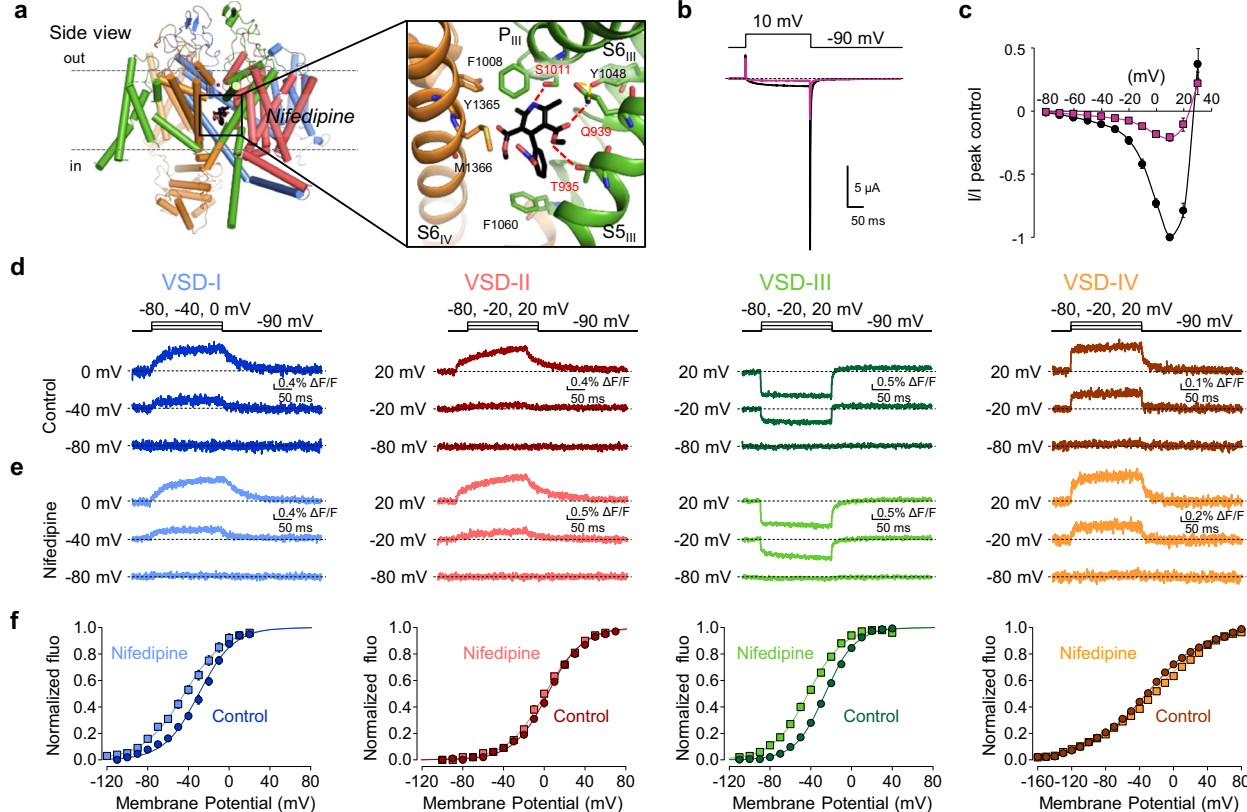

**Fig. 4 | Nifedipine selectively perturbs the activation of VSD-I and -III.**
**a** Structure of Ca$_V$1.1 $\alpha_{1S}$ interacting with nifedipine. Color code: repeat I in blue; repeat II in red; repeat III in green; repeat IV in yellow; Ca$^{2+}$ ions in purple and nifedipine in black. The inset shows an enlargement of the nifedipine docking site, which includes parts of the $\alpha_{1S}$ pore domain: S5$_{III}$, S6$_{III}$, P-loop$_{III}$, and S6$_{IV}$ helices. Side chains of putative residues coordinating nifedipine are shown. Red dashed lines represent potential hydrogen bonds (Protein Data Bank ID 6JP5, adapted from ref. [23]). **b** Representative Ba$^{2+}$ current traces before (black) and after 10 μM nifedipine (violet). **c** Normalized current-voltage relationship constructed from traces

as in (**b**) ($n = 6$). **d, e** Ca$_V$1.1 VSD activations in the absence ("control", **d**) and in the presence of 10 μM nifedipine (**e**). The voltage-clamp protocol is shown above the traces in panel (**d**). **f** F(V) curves for the four VSDs in the absence (circles) and presence of nifedipine (10 μM, squares). Note that nifedipine caused a leftward shift of the activation curves of VSD-I and -III, while leaving VSD-II and -IV unaffected. Data were fitted to the Boltzmann distribution. Fitting parameters are reported in Supplementary Table 3. Error bars are ± SEM (VSD-I: control $n = 11$, nifedipine $n = 5$; VSD-II: control $n = 8$, nifedipine $n = 5$; VSD-III: control $n = 14$, nifedipine $n = 14$; VSD-IV: control $n = 7$, nifedipine $n = 7$).

Rios and Brum[7] established that the charge movement originated from the Ca$_V$1.1 channel, which functions as a voltage sensor of Ca$^{2+}$ release. We now know that Ca$_V$1.1 comprises four distinct and independent VSDs (Fig. 1) with at least one of them functioning as RyR1 voltage sensors.

In our study, we have probed the properties of each VSD of the human Ca$_V$1.1 and singled out VSD-III as the only sensor with (i) kinetics, (ii) voltage dependence and (iii) pharmacological properties compatible with the properties of SR Ca$^{2+}$-release in skeletal muscle fibers (Figs. 2–4 and 5a). During an AP, the activation of VSD-III represents the earliest molecular transition responsible for the mechanical opening of RyR1 (Fig. 5b), thus coupling excitation with contraction.

**Two distinct Ca$_V$1.1 VSDs control two different ion channels**
Our recent work has revealed a stunning heterogeneity in the properties and functional role of the four VSDs of L- and N-type channels including Ca$_V$1.1, Ca$_V$1.2, Ca$_V$2.1, and Ca$_V$2.2[10,40,46–48]. However, Ca$_V$1.1 is unique among the Ca$_V$ family as its VSDs must control two separate processes with drastically different kinetics and voltage dependence: (i) opening of the Ca$_V$1.1 pore and (ii) opening of RyR1. Ca$_V$1.1 has assigned these two functions to two distinct VSDs: VSD-I, a slowly activating sensor, drives the opening of Ca$_V$1.1[10]. This sensor is the most energetically coupled to the pore, its activation contributing ~80 meV towards the stabilization of the open state[10], setting the ionic current

kinetics[10,49]. On the other hand, VSD-III controls RyR1, which inherits its fast kinetics and voltage dependence, distinctive of muscle SR Ca$^{2+}$ release (Figs. 2, 3). Interestingly, VSD-III has no role in gating Ca$_V$1.1 as it was found to have no contribution to Ca$_V$1.1 opening[10]. Thus, Ca$_V$1.1 VSD-III's exclusive role is to be the voltage sensor, in *trans*, of RyR1.

Prior work, that pioneered the use of voltage-clamp fluorometry in muscle fibers[11], could not conclusively assign a voltage-sensing role to a specific VSD(s) but found that the activation of VSD-III was slower than that of Ca$^{2+}$ release. However, in the present work, COVG-VCF provided the voltage control necessary to correctly assign the activities of all VSDs. Our positive identification of a single VSD consistent with a very stringent set of conditions (and positive exclusion of the other VSDs), including the voltage dependence of Ca$^{2+}$ release reported in several studies, and a muscle AP clamp, supports the validity of our findings and the major role of VSD-III in the EC coupling. Our finding is also consistent with the finding that mutations in VSD-III alone impaired the voltage dependence of Ca$^{2+}$ release, reported in a recent study on cultured murine myotubes[12].

An important structural element found to be essential for skeletal muscle EC coupling, is Ca$_V$1.1 "II-III loop" connecting Repeat-II and -III intracellularly (Fig. 1)[50,51]. The II-III loop is crucial for the propagation of Ca$_V$1.1 conformational rearrangements to RyR1, and perturbations in its amino acid sequence have been found to impair EC coupling[51]. While available structures of Ca$_V$1.1 alone[23,52,53] or in complex with RyR1[4] have not resolved this flexible region, several studies have highlighted

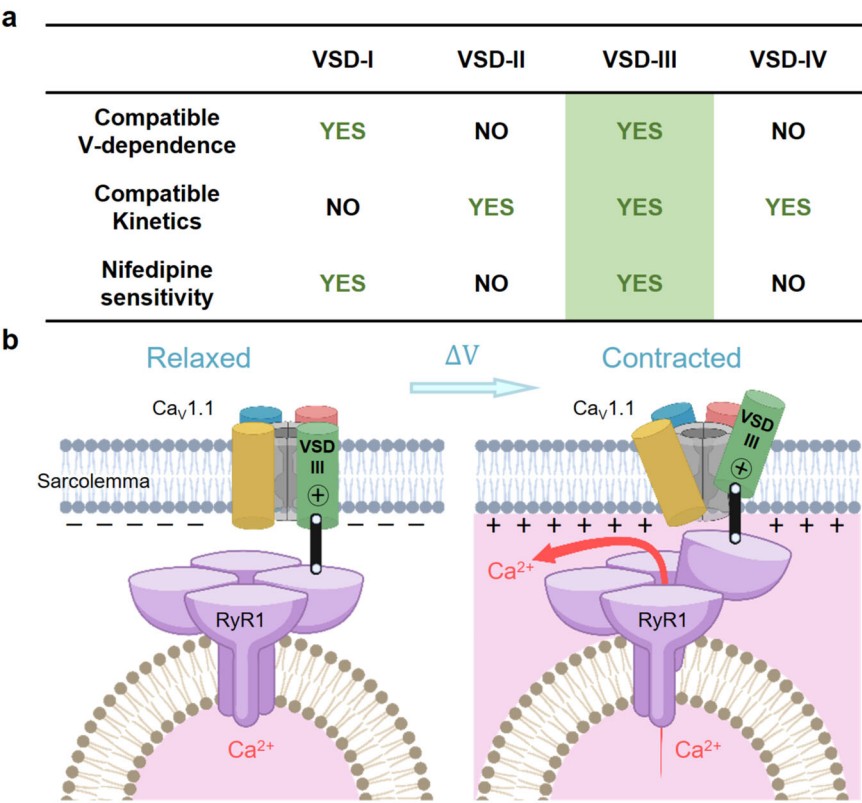

|  | VSD-I | VSD-II | VSD-III | VSD-IV |
|---|---|---|---|---|
| **Compatible V-dependence** | YES | NO | YES | NO |
| **Compatible Kinetics** | NO | YES | YES | YES |
| **Nifedipine sensitivity** | YES | NO | YES | NO |

**Fig. 5 | The activation of Ca_V1.1 VSD-III is the earliest molecular transition responsible for the electro-mechanical opening of RyR1. a** Summary of the VSD properties that fit those of skeletal muscle Ca$^{2+}$ release. **b** During a skeletal muscle AP, the displacement of the charged VSD-III controls RyR1 activation and muscle contraction. For simplicity, the interaction of one Ca_V1.1 and one RyR1 monomer is shown. Created in BioRender. Angelini, M. (2025) https://BioRender.com/z78y343.

a distinct region of interaction with RyR1[54]. Notably, VSD-III is the sensor directly connected with the II-III loop, which is likely to mediate the propagation of VSD-III voltage-dependent movements to RyR1, conferring voltage dependence to contraction. In addition, the β1a subunit[42] and Stac3 (which binds II-III loop[55,56]) have also been found necessary for muscle EC coupling[43,44].

## Nifedipine perturbs both muscle EC coupling and VSD-III

The role of VSD-III in EC coupling is also supported by its sensitivity to nifedipine, shared only with VSD-I (Fig. 4). Although skeletal muscle EC coupling does not depend on Ca$^{2+}$ influx[24,25], a surprisingly large number of studies found that nifedipine alters skeletal muscle contraction. Most of these studies report an overall reduction in Ca$^{2+}$ release[7], contraction[28,29,31–35,37], and force[27,38] in fibers and muscles. Other studies have observed the stimulatory effect of nifedipine[25,26,36,39]. While the endpoints of nifedipine action are still being debated, there is strong consensus on its ability to modify EC coupling, demonstrated in a variety of experimental settings and model species. Based on this body of work, it is not surprising to find that nifedipine altered VSD-III, the only sensor with kinetics and voltage dependence of SR Ca$^{2+}$ release.

Nifedipine binding to Ca_V1.1 channel induces a significant change in VSD-III activation by shifting its activation towards more negative membrane potentials (Fig. 4d–f), likely to be the perturbation responsible for altering EC coupling.

Interestingly, we found that nifedipine had also an effect on VSD-I activation (Fig. 4d–f). As nifedipine is known to stabilize an inactivated state of L-type Ca$^{2+}$ channels[20,57,58], the new conformational state of the nifedipine-bound channel is revealed as a leftward shift of the voltage dependency of VSD-I and -III (Fig. 4f). In the nifedipine-bound state, VSD-I and VSD-III appear uncoupled from the Ca_V1.1 pore and RyR1, as they still respond to depolarizations but are unable to gate Ca_V1.1 pore and RyR1, respectively, possibly explaining a reduction in EC coupling observed in most studies[7,27–29,31–35,37,38].

## Limitations and advantages of the *Xenopus* oocyte expression system in this study

A limitation of this study is that the conformational changes of the four VSDs have been captured in the absence of RyR1. We have attempted to reconstitute the EC coupling machinery in oocytes by expressing RyR1 together with the full Ca_V1.1 complex and Junctophilin, but we could not obtain evidence that functional coupling between the two channels occurred. Thus, Ca_V1.1 VSD activities are measured in a non-skeletal-muscle environment, independently of Ca$^{2+}$ release and any retrograde effect of RyR1 on Ca_V1.1. On the other hand, the *Xenopus* oocyte expression system, combined with the COVG-VCF technique[13,59,60] has provided unprecedented information on the biophysical properties of the individual VSDs of the human Ca_V1.1 channel, revealing their kinetics, voltage dependence, and sensitivity to nifedipine. The large signal-to-noise ratio of current and fluorescence of the oocyte expression system, combined with the fastest oocyte voltage-clamp (COVG), have revealed significant differences among the four VSDs, and singled out VSD-III as the voltage sensor of muscle contraction.

In summary, we have probed the excitation-driven molecular events within the Ca_V1.1 channel associated with EC coupling. Among the four VSDs, the activation of VSD-III is likely the earliest voltage-dependent molecular transition in the skeletal muscle EC coupling process. The other sensors do not possess fast kinetics (VSD-I), voltage dependence (VSD-II and -IV), or nifedipine sensitivity (VSD-II and -IV) that characterize skeletal muscle SR Ca$^{2+}$ release.

We propose that the conformational rearrangements of VSD-III propagate to RyR1, bestowing voltage sensitivity to this receptor: VSD-

III activation is the molecular event that makes muscle contraction an electrical phenomenon, as first observed in a frog leg by Luigi Galvani over two centuries ago[41].

## Methods

### Ethical statement
All animals are hosted in UCLA-maintained housing facilities; all animal procedures were approved by the UCLA Institutional Animal Care and Use Committee and conformed to the Guide for the Care and Use of Laboratory Animals published by the U.S. National Institutes of Health. Protocol numbers: ARC-2001-124 (*Xenopus laevis*) and ARC-2014-081-AM-003 (Mouse).

### Molecular biology
Human *CACNA1S* ($\alpha_{1S}$, GenBank accession no. BC133671)[61] was coexpressed with auxiliary subunits rabbit $\beta_{1a}$ (UniProt accession no. P19517), rabbit $\alpha_2\delta$-1 (UniProt accession no. P13806), human γ1 (UniProt accession no. Q06432) and mouse Stac3 (UniProt accession no. Q8BZ71) in *Xenopus laevis* oocytes.

A Cys was substituted in $\alpha_{1S}$ at the extracellular flank of the S4 helix of each VSD by site-directed mutagenesis, using the QuikChange Site-Directed Mutagenesis Kit (Agilent Technologies) to create L159C and L164C (VSD-I), M519C (VSD-II), V893C (VSD-III), or S1231C (VSD-IV) as previously described[10]. The cRNA of the different subunits was transcribed in vitro using mMESSAGE mMACHINE T7 (Ambion) or AmpliCap-Max T7 High Yield Message Maker Kit (CELLSCRIPT) and injected into *Xenopus laevis* oocytes.

### Oocyte preparation
Animals were sourced from Xenopus1, Corp USA. Oocyte lobes were surgically harvested from *Xenopus laevis* and defolliculated using collagenase type I (207 U/ml), as previously described[10]. Stage V-VI oocytes were injected at the equator with 50 nl cRNA mix containing the Ca$_V$1.1 complex.

Oocytes were incubated for 4–5 days at 18 °C in either SOS (in mM: 100 NaCl, 2 KCl, 1.8 CaCl$_2$, 1 MgCl$_2$, and 5 HEPES, pH = 7.0, with 100 U/ml penicillin, 100 μg/ml streptomycin, and 50 μg/ml gentamicin) or in a solution composed by 50% L-15 Leibovitz's L-15 (Corning cellgro), 47.5% H$_2$O, 10% heat-inactivated horse serum (HyClone), 100 U/ml penicillin, 100 μg/ml streptomycin, and 100 μg/ml amikacin (Cayman Chemical Company).

### Cut-open Oocyte Vaseline Gap - Voltage-Clamp Fluorometry (COVG-VCF)
4–5 days after injection, oocytes were incubated on ice with with thiol-reactive fluorophores, sensitive to environmental changes (10 μM tetramethylrhodamine-6-maleimide [TMRM-6′] (AAT Bioquest) for L159C or 10 μM tetramethylrhodamine-5-maleimide [TMRM-5′] (AAT Bioquest) for L164C or 20 μM MTS-5(6)-carboxytetramethylrhodamine [MTS-TAMRA] (Santa Cruz) for VSD-II, -III, or -IV) in a depolarizing solution (in mM: 120 K-methanesulfonate(MES), 2 Ba(MES)$_2$ or 2 Ca(MES)$_2$, and 10 HEPES, pH = 7.0). All stocks were 100 mM in DMSO.

Voltage-clamp fluorometry (VCF)[62] was performed at room temperature using the Cut-open oocyte Vaseline gap (COVG), implemented for epifluorescence measurements[13,59,60,63]. The VCF setup consists of an Olympus BX51WI upright microscope with an LED light source and appropriate filter set. Fluorescence emission was acquired using a 40× water immersion objective (LUMPlanFl, Olympus Optical) and amplified using a Dagan Photomax 200 system (Dagan Corporation).

External solution (mM) contained 120 NaMES, 2 Ca(MES)$_2$, 10 HEPES (pH = 7.0). Internal solution (mM) contained 120 K-Glutamate, 10 HEPES (pH = 7.0). Intracellular micropipette solution (mM) was composed of 2700 NaMES, 10 NaCl, and 10 HEPES (pH = 7.0). Prior to the experiments, oocytes were injected with 100 nl 100 mM BAPTA•4 K (Invitrogen), 10 mM HEPES, pH = 7.0, to prevent activation of native Ca$^{2+}$- and Ba$^{2+}$-dependent Cl$^-$ channels[64]. Nifedipine experiments were performed in 2 mM Ba(MES)$_2$ (Fig. 4), and L159C cysteine mutant was used for VSD-I VCF recordings (instead of L164C) since it provides a larger signal-to-noise ratio with a modest change in the voltage-dependent properties (Supplementary Figs. 1, 2 and Supplementary Table 1). Nifedipine (Alomone Labs) was dissolved in DMSO to make a 100 mM stock solution.

Fluorescence changes and ionic currents were elicited during a 200 ms depolarizing square pulse or using a skeletal muscle action potential (AP) waveform (see next section) as a voltage command. Holding potential was −90 mV. Ionic current and fluorescence were acquired simultaneously from the same membrane area and signals were filtered at 1/5 of the sampling frequency (typically 1–5 kHz).

### Skeletal muscle action potential
Animals were sourced from the Jackson lab, USA. Action potentials (APs) were recorded using the two-electrode voltage-clamp technique from enzymatically dissociated flexor digitorum brevis fibers from C57BL mice as previously described[65]. When filled with a solution mimicking the intracellular milieu ("intracellular solution"), both electrodes had resistances of 10-12 MΩ. The intracellular solution contained (in mM) 75 aspartate, 5 ATP-Na$_2$, 5 phospho-creatine di-Tris, 5 reduced glutathione, 5 MgCl$_2$, 30 EGTA, 15 Ca(OH)$_2$, and 20 MOPS, pH = 7.4 with KOH. The extracellular solution contained (in mM) 145 NaCl, 4 KCl, 2 CaCl$_2$, 1 MgCl$_2$, and 10 MOPS, 10 glucose, pH = 7.4 with NaOH. APs were elicited by supra-maximal 0.5 ms current pulses. AP waveform was digitized and used as a voltage command in VCF experiments.

### Data analysis
The voltage dependence of ionic conductance G(V), estimated from the peaks of the tail currents, was fitted to a Boltzmann equation:

$$G(V) = \frac{I_{tail,\,max}}{1 + \exp\left[zF(V_{1/2} - Vm)/(RT)\right]} \tag{1}$$

where $I_{tail,\,max}$ is the maximum tail current, $z$ is the valence, $V_{1/2}$ is the half-activation potential, Vm is the membrane potential, $F$ and $R$ are the Faraday and gas constants, and $T$ the absolute temperature, respectively. G(V) data points and fitting curves were normalized to the fitting parameter $I_{tail,\,max}$.

Fluorescence *vs* voltage curves (F(V)) were constructed by plotting the fluorescence amplitude at the end of 200 ms depolarizing pulses against the membrane potential. Data points were fitted to a Boltzmann equation in this form:

$$F(V) = \frac{F\,max - F\,min}{1 + \exp\left[zF(V_{1/2} - Vm)/(RT)\right]} + F_{min} \tag{2}$$

where Fmax is the maximum fluorescence deflection and Fmin is the minimum fluorescence deflection.

Fluorescence recordings under AP-clamp (Fig. 2b, d, e) were normalized as:

$$Norm\,F = \frac{F - F\,min}{F\,max - F\,min} \tag{3}$$

where F is the fluorescence recorded during the action potential, Fmin and Fmax are the minimum and maximal fluorescence, respectively, as determined during 200 ms depolarizing square pulse protocol (F(V)) in the same cell.

Fluorescence kinetics in Fig. 2a, c were fitted to one or the sum of two exponential function(s).

$$f(t) = B + \sum_{i=1}^{2} A_i \cdot \exp(-t/\tau i) \qquad (4)$$

where B is the baseline, A is the amplitude, t is time, and $\tau$ is the time constant.

## Statistical Analysis

Statistical significance was assessed using two-tailed unpaired Student's $t$-tests. Data are presented as means ± SEM.

## Protein structure depiction

Ca$_V$1.1 (Protein Data Bank ID 5GJV[53] and 6JP5[23]) and RyR1 (3J8H[66]) were rendered using PyMOL (Schrödinger). Supplementary Movie 1 was rendered in Blender 4.3 (the Blender Foundation) as previously for Ca$_V$2.2[47]. For the pore opening (conductance) in Supplementary Movie 1: first, the current trace recorded during the AP-clamp had the gating current subtracted. The ionic current trace was converted to conductance by dividing by the driving force using the reversal potential value calculated by a current-voltage plot from the same cell. The conductance trace was then normalized by dividing by the maximal conductance. This was derived from the cell's $I_{tail,max}$ divided by the driving force of the tail current.

## Reporting summary

Further information on research design is available in the Nature Portfolio Reporting Summary linked to this article.

## Data availability

The data that support this study are available from the corresponding authors upon request. Previously published structures and related PDB codes used in the manuscript are: 5GJV, 3J8H and 6JP5. Source data are provided in this paper.

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

## Acknowledgements

We are thankful to Drs. Michela Ottolia, Alan Neely, and Andreas Schwingshackl, and their laboratory members, for the insightful discussions and comments. This study was supported by grants from the National Institutes of Health/National Institute of General Medical Sciences R35GM131896 (R.O.), National Institute of Arthritis and Musculoskeletal and Skin Diseases R01AR063182 (S.C.), and R21AR080282-02R (M.D.). Start-up funds from the Linköping University Wallenberg Center for Molecular Medicine / the Knut and Alice Wallenberg Foundation (A.P.); Hjärnfonden (The Swedish Brain Foundation) grant FO2023-0025 (A.P.); Vetenskapsrådet (The Swedish Research Council) grant 2022-00574 (A.P.).

## Author contributions

Conceptualization: M.A., N.S., M.D., S.C., A.P., and R.O. Investigation: M.A., N.S., F.S., S.M., M.D., S.C., and R.O. Formal analysis M.A., N.S., F.S., S.P., A.P., and R.O. Methodology: M.A., N.S., F.S., A.P., and R.O. Funding acquisition: M.D., S.C., A.P., and R.O. Supervision: M.A. and R.O. Writing – original draft: M.A. and R.O. Writing – review & editing: All.

## Competing interests

The authors declare no competing interests.
