## [Transparent Peer Review file · Nature Communications]

The Molecular Transition that Confers Voltage Dependence to Muscle Contraction

Corresponding Author: Professor Riccardo Olcese

Version 0:

Reviewer comments:

Reviewer #1

(Remarks to the Author)

Angelini et al. describe experiments that investigate the relative roles of the four voltage sensor domains (VSDs) of CaV1.1 (VSD I, VSD II, VSD III, and VSD IV) in triggering muscle contraction. To analyze this, the authors employed the elegant Cut-open Oocyte Voltage-Clamp Fluorometry (COVC-VCF) technique, which they first developed in 2014 (PMID: 25489110), to assess the contribution of the four voltage sensors in CaV1.2 channel opening. The current findings build on the authors' prior work (PMID: 34546289), in which they examined the role of the four VSDs of CaV1.1 in channel activation. In Savalli et al., the authors concluded that the slow and right-shifted activation of CaV1.1 is primarily governed by VSDI, whereas VSDII, VSDIII, and VSDIV display faster kinetics more in line with the rapid calcium release from the sarcoplasmic reticulum (SR). In the current study, they extended their analysis by examining the activation kinetics of each VSD and their ability to respond to a single action potential. This allowed them to compare the kinetics and voltage dependence of VSD activation with those of SR calcium release in skeletal muscle (measured in other studies). Additionally, the authors explored the effect of nifedipine, a calcium channel blocker that disrupts CaV1.1 charge movement and, consequently, SR calcium release. Based on their findings, summarized in Figure 5, they conclude that VSDIII is the only VSD that exhibits both a voltage dependence and fast kinetics similar to those of SR calcium release. Furthermore, VSDIII is one of the two VSDs responsive to nifedipine, reinforcing their conclusion.

One weakness of the study is that its conclusion relies on a correlation between results obtained in oocytes—conditions that are optimal for COVC-VCF—but not directly relevant for EC coupling, and findings from several other studies conducted in muscle cells. Without the support of a recent study in myotubes, which utilized mutagenesis and molecular dynamics simulations to identify VSDIII as the key mediator of voltage-dependent SR calcium release (PMID: 39198449), the conclusion would be considerably weakened. Additionally, the manuscript would benefit from addressing the following points for further clarity and depth:

1. VSD I is described as 'the VSD most energetically coupled to the pore' and 'the one that drives the opening of CaV1.1' (lines 149 and 184). However, these conclusions are based on data from Savalli et al. (PMID: 34546289), in which the CaV1.1 $\alpha 1S$ subunit was co-expressed only with $\beta 1a$ and Stac3. In the current study, two additional accessory subunits, $\gamma 1$ and $\alpha 2\delta -1$, are co-transfected, resulting in a significant ~ 30 mV left-shift in the voltage dependence of VSD I, while the voltage dependence of the pore remains unaffected. This discrepancy suggests that $\alpha 2\delta -1$ may modulate the contribution of VSD I to pore opening, similar to what shown for CaV1.2 (PMID: 27481713). Therefore, the assumption that VSD I drives pore opening should be further examined within the context of this work, particularly by applying allosteric models to the presented data.
2. Nifedipine has been shown to occupy the fenestration between the pore domains of repeats III and IV in the CaV1.1 channel, as demonstrated in cryo-EM studies (PMID: 31150622). Based on this, one would expect that nifedipine would primarily affect the voltage-dependent conformational changes of voltage sensor domains (VSD) III and IV. However, the data presented here indicate that nifedipine influences VSD I and VSD III, but not VSD IV. To better understand the underlying mechanism of nifedipine's effect, an analysis of the energetic contributions to pore opening may provide further insight into how nifedipine modulates the voltage dependence of VSDs I, III, and IV.
3. To apply the COVG-VCF technique the authors need to introduce cysteine mutations. For VSDI, they previously used L159C. Here in supplementary figure 1, they show both L159C and L164C, which seems to have been used in the current paper. However, no rationale is given on to why L159C was not suitable for the current study and why L164C was preferable.
4. In figure 2, it is indicated that the CaV1.1 complex consists of $\alpha 1s$, $\beta 1a$, $\alpha 2d -1a$ and stac3. What is meant with $\alpha 2d -1a$? Is it referring to the NCBI annotation or is it a typo?
5. At line 64, COVG-VCF for Cut-open Oocyte Voltage-Clamp Fluorometry, does VG stand for voltage-gated?

Reviewer #2

(Remarks to the Author)

Reviewer #3

(Remarks to the Author)

The main result of this study and the noteworthy result is that Marina and colleagues provide additional evidence in support of the role of the voltage sensing domain III (VSD-III) of CaV1.1 in skeletal muscle EC coupling. The contribution of VSD-III has been implicated in a number of previous studies including those of Banks et al 2001 and Pelizzari et al 2024. In addition, the importance of the II-III loop in EC coupling makes it very likely that the nearby VSD-III would be involved. Never-the less the present article, although not surprising, does provide evidence that VSD-III may be in EC coupling. Therefore, the result will be of significance to the field. A negative aspect is that all results were obtained in oocytes where Ca²⁺ release cannot be measured. The manuscript would be much stronger if parallel results obtained using a skeletal muscle system such as myotubes, where Ca²⁺ release from sarcoplasmic reticulum could be measured at the same time as VSD responses.

The work does support the conclusions, the data analysis is appropriate, the methodology sound and meets expected standards in the field. The detail in the Methods is adequate.

One incorrect statement in Line 204 is that structure of CaV1.1 in complex with RyR1 are low resolution and therefore NOT "atomic" structures.

A further misleading statement is line 61-62. "VSD activation and Ca²⁺ release have shared pharmacology i.e. the VSD is modified by compounds known to modify Ca²⁺ release." Many compounds could modify Ca²⁺ release by interacting directly with RyR1, STAC3, or the Cav1.1 beta subunit and thus modify Ca²⁺ release without necessarily affecting the VSD or could affect the VSD through a non-specific interaction or via retrograde signaling. This statement needs to be rewritten.

Version 1:

Reviewer comments:

Reviewer #1

(Remarks to the Author)

The authors have addressed several of the criticisms raised in the first round of review by discussing the points raised and providing a preliminary allosteric CaV1.1 model to predict the effects of $\gamma 1$ and $\alpha 2\delta 1$. However, I would like the authors to clarify a couple of points that may improve the understanding of the manuscript.

The analysis of the R174W mutation in Savalli et al. JGP 2021, and of the inclusion of $\gamma 1$ and $\alpha 2\delta 1$ here reported make a strong case that the VSD I is the most energetically coupled to the pore. However, the 30 mV right shift in the V_{1/2} of VSD I observed in R174W corresponded to a similar 35 mV right shift in the V_{1/2} of pore opening. Here, instead, while the inclusion of $\gamma 1$ and $\alpha 2\delta 1$ induced a 30 mV left shift in the V_{1/2} of VSD I, along with an enhancement of the VSD coupling energy (from -81 meV to -99 meV), making VSD I even more dominant in the regulation of pore opening, the pore opening displayed a mere 10 mV left shift. This mismatch between VSD I and pore shift of the voltage dependence would suggest (to me) a reduction in the VSD coupling energy rather than an enhancement. How do the authors reconcile these differences?

The result of the model prediction for pore opening shown in Table R1C indicates a V_{1/2} of the pore at 88mV for both conditions, with or without $\gamma 1$ and $\alpha 2\delta 1$. I assume this value derives from the analysis of the tail currents. However, the experimental data suggest a 10 mV right shift in the V_{1/2} of the currents between the two conditions. Therefore, why does the model not account for this experimental shift? Could the discrepancy mentioned in the point above arise from the assumption that there is no difference in the V_{1/2} of pore opening (reported as 88 mV for both conditions)? Alternatively, could a more thorough analysis help clarify this conundrum?

Reviewer #2

(Remarks to the Author)

Reviewer #3

(Remarks to the Author)

The authors have adequately addressed two of my concerns but not the third concern that the results were obtained in

oocytes where Ca²⁺ release cannot be measured. Although the point is now addressed, the concern still remains that the experiments were not done in cells related to muscle. Therefore, the Title “The Molecular Transition that Confers Voltage Dependence to Muscle Contraction” is misleading because no signal directly related to muscle contraction has been measured. For the title not to be misleading it would have to be changed to something like “A Molecular Transition that Potentially Confers the Voltage Dependence of Muscle Contraction”. Also the wording of some of the related conclusion also needs to be changed to support the new title.

NCOMMS-24-63826 RESPONSE TO REVIEWER COMMENTS

We sincerely thank the Reviewers for the positive appraisal of our work and constructive comments. We have addressed all the concerns and made changes to the revised manuscript accordingly. We have included a new Supplementary Figure 2 describing the two cysteine mutants used for VCF experiments of VSD-I. In addition, to better illustrate our results, we have prepared an animation where the Ca_v1.1 structure is annotated to show the extent of VSD activation and pore opening during a muscle action potential (new Supplementary Movie 1). We have added an author, Serena Pozzi, who contributed Supplementary Movie 1.

Major changes in the manuscript are in blue color fonts.

Please find below our point-by-point answers to Reviewers' comments quoted in blue font, *Italic*.

Reviewer #1 (Remarks to the Author):

“Angelini et al. describe experiments that investigate the relative roles of the four voltage sensor domains (VSDs) of Ca_v1.1 (VSD I, VSD II, VSD III, and VSD IV) in triggering muscle contraction. To analyze this, the authors employed the elegant Cut-open Oocyte Voltage-Clamp Fluorometry (COVC-VCF) technique, which they first developed in 2014 (PMID: 25489110), to assess the contribution of the four voltage sensors in Ca_v1.2 channel opening. The current findings build on the authors' prior work (PMID: 34546289), in which they examined the role of the four VSDs of Ca_v1.1 in channel activation. In Savalli et al, the authors concluded that the slow and right-shifted activation of Ca_v1.1 is primarily governed by VSDI, whereas VSDII, VSDIII, and VSDIV display faster kinetics more in line with the rapid calcium release from the sarcoplasmic reticulum (SR). In the current study, they extended their analysis by examining the activation kinetics of each VSD and their ability to respond to a single action potential. This allowed them to compare the kinetics and voltage dependence of VSD activation with those of SR calcium release in skeletal muscle (measured in other studies). Additionally, the authors explored the effect of nifedipine, a calcium channel blocker that disrupts Ca_v1.1 charge movement and, consequently, SR calcium release. Based on their findings, summarized in Figure 5, they conclude that VSDIII is the only VSD that exhibits both a voltage dependence and fast kinetics similar to those of SR calcium release. Furthermore, VSDIII is one of the two VSDs responsive to nifedipine, reinforcing their conclusion

One weakness of the study is that its conclusion relies on a correlation between results obtained in oocytes—conditions that are optimal for COVC-VCF—but not directly relevant for EC coupling, and findings from several other studies conducted in muscle cells.

We sincerely thank the Reviewer for the constructive suggestions and agree with their comment regarding the absence of SR Ca²⁺ release: in the revised version, we have discussed limitations and advantages of our approach (*Xenopus* oocyte expression system and COVG-VCF). The following new paragraph is now included in the Discussion section:

“Limitations and advantages of the *Xenopus* oocyte expression system in this study

A limitation of this study is that the conformational changes of the four VSDs have been captured in the absence of RyR1. We have attempted to reconstitute the EC coupling machinery in oocytes by expressing RyR1 together with the full Ca_v1.1 complex and Junctophilin, but we could not obtain evidence that functional coupling between the two channels occurred. Thus, Ca_v1.1 VSD activities are measured

independently of Ca^{2+} release and any retrograde effect of RyR1 on $\text{Ca}_v1.1$. On the other hand, the *Xenopus* oocyte expression system, combined with the COVG-VCF technique¹⁻³ has provided unprecedented information on the biophysical properties of the individual VSDs of the human $\text{Ca}_v1.1$ channel, revealing their kinetics, voltage-dependence and sensitivity to nifedipine. The large signal-to-noise ratio of current and fluorescence of the oocyte expression system, combined with the fastest oocyte voltage-clamp (COVG), have revealed significant differences among the four VSDs, and singled out VSD-III as the voltage sensor of muscle contraction.”

Without the support of a recent study in myotubes, which utilized mutagenesis and molecular dynamics simulations to identify VSDIII as the key mediator of voltage-dependent SR calcium release (PMID: 39198449), the conclusion would be considerably weakened”.

We respectfully disagree with this comment. The myotube study provided a more physiological context than our preparation; our study provided a direct observation of VSD function, free of the assumptions carried by mutagenesis and molecular dynamics simulations. It is highly encouraging that the two independent studies reached the same conclusion, and both are supported by each-other’s findings.

Additionally, the manuscript would benefit from addressing the following points for further clarity and depth:

1. VSD I is described as 'the VSD most energetically coupled to the pore' and 'the one that drives the opening of $\text{Ca}_v1.1$ ' (lines 149 and 184). However, these conclusions are based on data from Savalli et al. (PMID: 34546289), in which the $\text{Ca}_v1.1$ $\alpha_1\text{S}$ subunit was co-expressed only with $\beta_1\text{a}$ and Stac3 . In the current study, two additional accessory subunits, γ_1 and $\alpha_2\delta-1$, are co-transfected, resulting in a significant ~ 30 mV left-shift in the voltage dependence of VSD I, while the voltage dependence of the pore remains unaffected. This discrepancy suggests that $\alpha_2\delta-1$ may modulate the contribution of VSD I to pore opening, similar to what shown for $\text{Ca}_v1.2$ (PMID: 27481713). Therefore, the assumption that VSD I drives pore opening should be further examined within the context of this work, particularly by applying allosteric models to the presented data.

There is no discrepancy with our previous conclusions about the relevance of VSD-I on $\text{Ca}_v1.1$ activation, and we apologize for the lack of clarity regarding the effect of γ_1 and $\alpha_2\delta-1$ on the regulation of $\text{Ca}_v1.1$ voltage dependence. As the Reviewer expected, a shift in the voltage dependence of a sensor energetically coupled with the pore should also be associated with a change in the position of the G(V) curve. Indeed, this is what we have observed for the voltage-dependence of VSD-I and the G(V). Under identical ionic conditions (2 mM ext Ba^{2+}), the two additional auxiliary subunits (γ_1 , $\alpha_2\delta-1$) used in this new study caused a ~ 30 mV leftward-shift of VSD-I activation; this change is associated with a significant ~ 10 mV shift, in the same direction of the voltage dependence of channel opening. G(V) curve: **$\alpha_1\text{S}+\beta_1\text{a}+\text{Stac3}$** : $V_{1/2}=31 \pm 1$ mV (n=7) from Savalli et al. *JGP* 2021 (PMID: 34546289; <https://doi.org/10.1085/jgp.202112915>). **$\alpha_1\text{S}+\beta_1\text{a}+\text{Stac3}+\gamma_1+\alpha_2\delta-1$** : $V_{1/2}=20 \pm 1$ mV (n = 7, Supplementary Table 1, this work).

To avoid confusion, in the revised manuscript, we report the ionic conditions also in the legends of Supplementary Table 1-3. We have also added $V_{1/2}$ and z values of the G(V) curve for the WT channel in Supplementary Table 1, that were not included in the previous version of the manuscript.

The Reviewer's suggestion to further explore the energetic contribution of $\text{Ca}_v1.1$ VSDs to the pore opening in the presence of other auxiliary subunits is an excellent one and something we are currently working on. We have preliminarily interrogated the allosteric $\text{Ca}_v1.1$ model presented by Savalli et al. *JGP* 2021 to predict the steady state properties of the VSDs and pore opening in the presence of γ_1 and $\alpha_2\delta-1$. As shown in Figure R1 (prepared for the Reviewers), it is sufficient to change only VSD-I parameters (V_1 , q_1) and increase its interaction energy VSDs \leftrightarrow Pore (W_1) to adequately fit all the steady state curves in the presence of all auxiliary subunits (Figure R1 b-c). These data suggest that γ_1 and $\alpha_2\delta-1$ subunits favor

$\text{Ca}_v1.1$ voltage-dependent activation (more negative $G(V)$) by enhancing the allosteric coupling between VSD-I and pore opening (W_1 is more negative) (Figure R1C). However, a full computational study also involving the kinetics parameters of the current and VSDs is necessary to estimate and evaluate all possible solutions. We hope that the Reviewer agrees that the effect of γ_1 and $\alpha_2\delta-1$ on VSDs \leftrightarrow Pore energetics deserves a full, dedicated study, similarly to what we have presented in Savalli et al., *JGP* 2016 to understand the mechanism by which $\alpha_2\delta$ facilitates the opening of cardiac $\text{Ca}_v1.2$. This extensive effort is outside the scope of this work, as here we focused on the role of VSD activation for Ca^{2+} release (RyR1 opening), instead of $\text{Ca}_v1.1$ pore opening. Thank you for prompting us to clarify our work.

2. Nifedipine has been shown to occupy the fenestration between the pore domains of repeats III and IV in the $\text{Ca}_v1.1$ channel, as demonstrated in cryo-EM studies (PMID: 31150622). Based on this, one would expect that nifedipine would primarily affect the voltage-dependent conformational changes of voltage sensor domains (VSD) III and IV. However, the data presented here indicate that nifedipine influences VSD I and VSD III, but not VSD IV. To better understand the underlying mechanism of nifedipine's effect, an analysis of the energetic contributions to pore opening may provide further insight into how nifedipine modulates the voltage dependence of VSDs I, III, and IV.

These are interesting suggestions and exploring $\text{Ca}_v1.1$ block by nifedipine would add valuable information on the pharmacological mechanism of dihydropyridines inhibition. However, in the absence of ionic conductance, the allosteric model cannot evaluate the coupling energy of sensors with the pore.

Nevertheless, we can still conclude that in nifedipine-blocked channels, the activation of VSD-I and VSD-III are perturbed, revealing a new conformational state of the channel. As nifedipine is thought to favor an inactivated state of $Ca_v1.1$, one may speculate that VSD-I and -III are in the “relaxed” or “converted” state where their activation is facilitated but uncoupled from the pore. However, when we challenged $Ca_v1.1$ with prolonged depolarizations to drive the channel into the inactivated state, we observed that in addition to VSD-I and -III also VSD-II is “converted” (Olcese lab data, unpublished). Thus, nifedipine-block state does not fully recapitulate the features of the inactivated state. At this stage, we recognize that we do not know enough on how nifedipine blocks the channel, except that it perturbs VSD-I (the sensor we proposed to contribute to channel opening) and VSD-III; the latter offers an explanation as to why nifedipine alters SR Ca^{2+} release.

3. To apply the COVG-VCF technique the authors need to introduce cysteine mutations. For VSDI, they previously used L159C. Here in supplementary figure 1, they show both L159C and L164C, which seems to have been used in the current paper. However, no rationale is given as to why L159C was not suitable for the current study and why L164C was preferable.

We apologize for the lack of clarity on the use of the two cysteine mutants used for fluorescent labeling of VSD-I. As reported in the Methods section of the original submission, L159C provided a larger signal-to-noise ratio compared to L164C position, and L159C was used only to evaluate the effect of nifedipine that required experiments with longer duration. Since this construct has a slightly shallower G(V) curve compared to WT (see Supplementary Figure 1, light blue and Supplementary Table 1, both original submission and new submission) and a modest change in the voltage-dependent properties F(V) (Figure R2 in the response to the Reviewer Comments and new Supplementary Figure 2), L164C has been our preferred construct whenever possible.

We have updated the Method section and included a new Supplementary Figure 2.

4. In figure 2, it is indicated that the $Ca_v1.1$ complex consists of α_1s , $b1a$, α_2d-1a and $stac3$. What is meant with α_2d-1a ? Is it referring to the NCBI annotation or is it a typo?

Thank you: it is a typo, and we have corrected Figure 2.

5. At line 64, COVG-VCF for Cut-open Oocyte Voltage-Clamp Fluorometry, does VG stand for voltage-gated?

We thank the Reviewer for pointing out this mistake. COVG-VCF stands for Cut-open Oocyte Vaseline Gap - Voltage-Clamp Fluorometry. We have corrected the text accordingly.

Reviewer #2 (Remarks to the Author):

This is great initiative: thank you for co-reviewing our manuscript!

Reviewer #3 (Remarks to the Author):

The main result of this study and the noteworthy result is that Marina and colleagues provide additional evidence in support of the role of the voltage sensing domain III (VSD-III) of CaV1.1 in skeletal muscle EC coupling. The contribution of VSD-III has been implicated in a number of previous studies including those of Banks et al 2001 and Pelizzari et al 2024. In addition, the importance of the II-III loop in EC coupling makes it very likely that the nearby VSD-III would be involved. Never-the less the present article, although not surprising, does provide evidence that VSD-III may be in EC coupling. Therefore, the result will be of significance to the field. A negative aspect is that all results were obtained in oocytes where Ca²⁺ release cannot be measured. The manuscript would be much stronger if parallel results obtained using a skeletal muscle system such as myotubes, where Ca²⁺ release from sarcoplasmic reticulum could be measured at the same time as VSD responses.

The work does support the conclusions, the data analysis is appropriate, the methodology sound and meets expected standards in the field. The detail in the Methods is adequate.

We thank the Reviewer for the positive assessment of our study. We have addressed all the concerns and made changes to the manuscript accordingly. As the Reviewer noted, our study is conducted in the absence of SR Ca²⁺ release: in the revised version, we included a new paragraph in the Discussion section (page 12) to highlight “Limitations and advantages of the *Xenopus* oocyte expression system in this study” (please see also our response to Reviewer 1 on the same point).

One incorrect statement in Line 204 is that structure of CaV1.1 in complex with RyR1 are low resolution and therefore NOT “atomic” structures.

We agree: we have removed “atomic”.

A further misleading statement is line 61-62. “VSD activation and Ca²⁺ release have shared pharmacology i.e. the VSD is modified by compounds known to modify Ca²⁺ release.” Many compounds could modify Ca²⁺ release by interacting directly with RyR1, STAC3, or the Cav1.1 beta subunit and thus modify Ca²⁺ release without necessarily affecting the VSD or could affect the VSD through a non-specific interaction or via retrograde signaling. This statement needs to be rewritten.

We agree with the Reviewer: we have modified the sentence to clarify this point as follows:

“A drug that modifies the activation of the VSD(s) assigned to RyR1, should also affect Ca²⁺ release.”

References:

- 1 Stefani, E. & Bezanilla, F. Cut-open oocyte voltage-clamp technique. *Methods Enzymol* **293**, 300-318, doi:10.1016/s0076-6879(98)93020-8 (1998).
- 2 Gandhi, C. S. & Olcese, R. The voltage-clamp fluorometry technique. *Methods Mol Biol* **491**, 213-231, doi:10.1007/978-1-59745-526-8_17 (2008).

- 3 Pantazis, A. & Olcese, R. in *Encyclopedia of Biophysics* (eds Gordon Roberts & Anthony Watts) 1-9 (Springer Berlin Heidelberg, 2019).

NCOMMS-24-63826B RESPONSE TO REVIEWER COMMENTS

We are very thankful for the suggestions. Please find below our point-by-point answers to the Reviewers' comments, quoted in blue font, *Italic*.

Reviewer #1 (Remarks to the Author):

The authors have addressed several of the criticisms raised in the first round of review by discussing the points raised and providing a preliminary allosteric CaV1.1 model to predict the effects of γ_1 and $\alpha_2\delta_1$. However, I would like the authors to clarify a couple of points that may improve the understanding of the manuscript.

- 1) The analysis of the R174W mutation in Savalli et al. JGP 2021, and of the inclusion of γ_1 and $\alpha_2\delta_1$ here reported make a strong case that the VSD I is the most energetically coupled to the pore. However, the 30 mV right shift in the V1/2 of VSD I observed in R174W corresponded to a similar 35 mV right shift in the V1/2 of pore opening. Here, instead, while the inclusion of γ_1 and $\alpha_2\delta_1$ induced a 30 mV left shift in the V1/2 of VSD I, along with an enhancement of the VSD coupling energy (from -81 meV to -99 meV), making VSD I even more dominant in the regulation of pore opening, the pore opening displayed a mere 10 mV left shift. This mismatch between VSDI and pore shift of the voltage dependence would suggest (to me) a reduction in the VSD coupling energy rather than an enhancement. How do the authors reconcile these differences?*
- 2) The result of the model prediction for pore opening shown in Table R1C indicates a V1/2 of the pore at 88mV for both conditions, with or without γ_1 and $\alpha_2\delta_1$. I assume this value derives from the analysis of the tail currents. However, the experimental data suggest a 10 mV right shift in the V1/2 of the currents between the two conditions. Therefore, why does the model not account for this experimental shift? Could the discrepancy mentioned in the point above arise from the assumption that there is no difference in the V1/2 of pore opening (reported as 88 mV for both conditions)? Alternatively, could a more thorough analysis help clarify this conundrum?*

These are two great points: as they are indeed related, for clarity, we shall address question #2 first.

The Ca_v channel allosteric model (first presented in Pantazis et al., *PNAS* 2014) describes the behaviors of five energetically coupled “particles”: four Voltage Sensing Domains and one central Pore domain. Each particle can exist in two states: Resting-Active (VSDs) and Open-Closed (Pore), thus allowing for $2^5 = 32$ possible states. Each VSD interacts with the Pore *via* an energetic factor W. The central assumption of the Ca_v allosteric model is that the pore has its own intrinsic voltage dependence with a very depolarized half-activation potential (**VL= +88 mV**) such that the pore effectively does not open under physiological depolarizations. However, when one or more VSDs are energetically coupled with the pore, their activation stabilizes the Open state of the pore (by an energetic factor W) “pulling” its voltage dependence to the left within a physiological range of V_m.

So, to clarify, “**VL**” is not the V_{half} estimated by fitting a Boltzmann curve to tail currents, but the intrinsic half-activation potential of the Pore.

In the simple model exercise presented in response to the previous comments of this Reviewer, we fixed the VL parameter to +88 mV, previously estimated for Cav1.1 without $\gamma_1 + \alpha_2\delta$ (Savalli et al., *JGP* 2021), making the assumption that the γ_1 and $\alpha_2\delta_1$ subunits do not affect the intrinsic properties of the pore.

Addressing now question #1, please note within the framework of Savalli et al. *JGP* 2021 allosteric model, a shift in the V_{half} of the VSD-I does not necessarily translate into an identical shift of the V_{half} of the G(V). In the case of R174W mentioned by the Reviewer, a simple rightward shift (+30mV) of VSD-I moves the GV by ~10 mV. To account for the larger (~+30mV) rightward shift of the G(V) caused by R174W it is necessary a concomitant reduction (~70%) in the coupling free energy ($W1$) of VSD-I \leftrightarrow Pore (from $W1_{\text{WT}} = -81\text{meV}$ to $W1_{\text{R174W}} = -22\text{meV}$). The reduction of the coupling free energy allows the GV to approach the more positive intrinsic voltage dependence (VL) of the pore.

Similarly, to account for the effect of $\gamma1+\alpha2\delta$, the simple model exercise prompted by the question of the Reviewer, suggests that these modulatory subunits alter the intrinsic voltage dependence of VSD-I and its coupling energy with the pore. That is, the hyperpolarizing shift in the voltage dependence of VSD-I alone is not sufficient to fully account for the leftward shift in the G(V) caused by $\gamma1+\alpha2\delta$ and an increase of the coupling energy (from $W1_{\text{WT}} = -81\text{meV}$ to $W1_{\text{with } \gamma1+\alpha2\delta} = -99\text{meV}$) is necessary to bring the G(V) curve closer to VSD-I curve and properly fit the experimental data. This exercise suggests that $\gamma1+\alpha2\delta$ increase the energetic coupling VSD-I \leftrightarrow Pore by ~ 20%, making VSD-I even more relevant to pore opening.

Again, while insightful, this is a preliminary analysis we performed to answer the question of the Reviewer, assuming that only VSD-I is perturbed by $\gamma1+\alpha2\delta$ (a reasonable assumption because, in the $\text{Ca}_v1.1$ CryoEM study (Wu et al., *Nature* 2016) $\alpha2\delta-1$ subunit directly interacts with the Repeat-I of the channel). As we mentioned before, to fully understand the energetics of $\text{Ca}_v1.1$ modulation by $\gamma1+\alpha2\delta$ would require an exhaustive study involving simultaneous fitting of steady-state and kinetic experimental data of $\text{Ca}_v1.1$ pore opening and VSDs activation, in different subunit compositions: this effort is outside the scope of our work, which aimed to identify the VSD(s) with biophysical properties consistent with Ca^{2+} release, and not relevant to the conclusions of the present study.

Reviewer #2 (Remarks to the Author):

Thank you for co-reviewing our manuscript!

Reviewer #3 (Remarks to the Author):

The authors have adequately addressed two of my concerns but not the third concern that the results were obtained in oocytes where Ca^{2+} release cannot be measured. Although the point is now addressed, the concern still remains that the experiments were not done in cells related to muscle. Therefore, the Title "The Molecular Transition that Confers Voltage Dependence to Muscle Contraction" is misleading because no signal directly related to muscle contraction has been measured. For the title not to be misleading it would have to be changed to something like "A Molecular Transition that Potentially Confers the Voltage Dependence of Muscle Contraction". Also the wording of some of the related conclusion also needs to be changed to support the new title.

Thank you for suggesting a new title. We ask this Reviewer to kindly reconsider their request, given the following:

- 1) All scientific work is only “potentially” true.
- 2) Throughout the paper, we make our experimental setting very clear, and take extra considerations to ensure that what we observe in the oocyte aligns with current knowledge on muscle Ca^{2+} release.
- 3) In the Introduction we clearly state the criteria for the molecular transition driving SR- Ca^{2+} release.
- 4) As this Reviewer previously requested, we added a new paragraph in the Discussion on the limitations and advantages of the oocyte system and approach used in our study.
- 5) Our conclusions are supported by the recent study in cultured myotubes, also published in this journal.

As other Reviewers did not raise concerns about the title, we would like to keep it as in the original submission. We do however take this Reviewer’s point, and we have softened the conclusion (page 12) as follows:

“Thus, $\text{Ca}_v1.1$ VSD activities are measured in a non-skeletal-muscle environment, independently of Ca^{2+} release and any retrograde effect of RyR1 on $\text{Ca}_v1.1$.”

“Among the four VSDs, the activation of VSD-III is likely the earliest voltage-dependent molecular transition in the skeletal muscle EC coupling process.”